# Metabolic Capacity of the Antarctic Cyanobacterium *Phormidium pseudopriestleyi* That Sustains Oxygenic Photosynthesis in the Presence of Hydrogen Sulfide

**DOI:** 10.3390/genes12030426

**Published:** 2021-03-16

**Authors:** Jessica E. Lumian, Anne D. Jungblut, Megan L. Dillion, Ian Hawes, Peter T. Doran, Tyler J. Mackey, Gregory J. Dick, Christen L. Grettenberger, Dawn Y. Sumner

**Affiliations:** 1Microbiology Graduate Group, University of California, Davis, CA 95616, USA; jemizzi@ucdavis.edu; 2Life Sciences Department, The Natural History Museum, London SW7 5BD, UK; a.jungblut@nhm.ac.uk; 3Genomics and Bioinformatics, Novozymes, Inc., Davis, CA 95616, USA; mled@novozymes.com; 4Coastal Marine Field Station, University of Waikato, Tauranga 3110, New Zealand; ian.hawes@waikato.ac.nz; 5Geology and Geophysics, Louisiana State University, Baton Rouge, LA 70803, USA; pdoran@lsu.edu; 6Department of Earth and Planetary Sciences, University of New Mexico, Albuquerque, NM 87131, USA; tjmackey@unm.edu; 7Department of Earth and Environmental Sciences, University of Michigan, Ann Arbor, MI 48109, USA; gdick@umich.edu; 8Department of Earth and Planetary Sciences, University of California, Davis, CA 95616, USA; clgrettenberger@ucdavis.edu

**Keywords:** cyanobacteria, cryosphere, genomics, sulfide, photosynthesis, lake, Antarctica

## Abstract

Sulfide inhibits oxygenic photosynthesis by blocking electron transfer between H_2_O and the oxygen-evolving complex in the D1 protein of Photosystem II. The ability of cyanobacteria to counter this effect has implications for understanding the productivity of benthic microbial mats in sulfidic environments throughout Earth history. In Lake Fryxell, Antarctica, the benthic, filamentous cyanobacterium *Phormidium pseudopriestleyi* creates a 1–2 mm thick layer of 50 µmol L^−1^ O_2_ in otherwise sulfidic water, demonstrating that it sustains oxygenic photosynthesis in the presence of sulfide. A metagenome-assembled genome of *P. pseudopriestleyi* indicates a genetic capacity for oxygenic photosynthesis, including multiple copies of *psbA* (encoding the D1 protein of Photosystem II), and anoxygenic photosynthesis with a copy of *sqr* (encoding the sulfide quinone reductase protein that oxidizes sulfide). The genomic content of *P. pseudopriestleyi* is consistent with sulfide tolerance mechanisms including increasing *psbA* expression or directly oxidizing sulfide with sulfide quinone reductase. However, the ability of the organism to reduce Photosystem I via sulfide quinone reductase while Photosystem II is sulfide-inhibited, thereby performing anoxygenic photosynthesis in the presence of sulfide, has yet to be demonstrated.

## 1. Introduction

Cyanobacterial production of O_2_ from oxygenic photosynthesis (OP) oxidized the Earth’s atmosphere during the Great Oxidation Event 2.4 billion years ago, which changed various elemental cycles, including the sulfur cycle [1]. Specifically, the Great Oxidation Event increased oxidative weathering, leading to a large flux of sulfate to the ocean, allowing more microbial sulfate reduction and increased sulfide concentrations in high productivity environments [2]. The biogeochemistry of OP is influenced by the presence of sulfide, which normally inhibits OP, presenting a challenge for cyanobacteria living in the presence of sulfide in diverse environments since the Great Oxidation Event. About 750 million years ago, a global “Snowball Earth” glaciation is associated with significant sulfate reduction [3,4,5], suggesting that cyanobacteria may have sustained OP in cold, sulfidic environments during at least one global glaciation. The only cold environment where sulfide-tolerant OP has been described is Middle Island Sinkhole in Michigan, USA (8–10 °C) [6], even though cyanobacteria are the dominant primary producers in many extreme cold ecosystems [7]. Sulfide-tolerant O_2_ production has also been documented in springs and sink holes in warmer environments, including Frasassi springs in Italy [8,9] and Little Salt Spring sinkhole in Florida, USA [10,11].

Prior work identified a cyanobacterium, *P. pseudopriestleyi* in ice-covered Lake Fryxell, Antarctica [12,13], that creates a thin layer of O_2_ in sulfidic pore water, an “oxygen oasis” [12,13]. Compared to other environments where sulfide-tolerant OP has been observed, Lake Fryxell is colder (2.4–2.7 °C), and its high latitude leads to months of continuous winter darkness. Based on sulfide and O_2_ fluxes, the oxygen oasis is expected to disappear during the dark winter months when photosynthesis does not occur. The increasing availability of light in spring allows benthic mats dominated by *P. pseudopriestleyi* to transition from sulfidic to oxic conditions due to photosynthetic O_2_ production [12]. However, the mechanism by which *P. pseudopriestleyi* tolerates sulfide and initiates this redox change is unknown.

Sulfide inhibits OP by blocking electron donation interaction between H_2_O and the Mn_4_CaO_5_ cluster at the oxygen-evolving complex (OEC) in the D1 protein of Photosystem II (PS II). Cyanobacteria respond to sulfide in one of four ways: (1) complete inhibition of OP, (2) continued but partial inhibition of OP, (3) simultaneous OP and anoxygenic photosynthesis (AP), or (4) shutting down of OP and use of AP until enough sulfide is oxidized that OP can start again [14,15,16] (Figure 1). Response 1 results in a cessation of the electron flow by blocking the OEC in PS II with no source of electrons for Photosystem I (PS I). Response 2 consists of the sulfide quinone reductase (SQR) protein oxidizing sulfide to elemental sulfur and providing electrons to PS I to perform AP. Response 3 involves modification of the D1 protein by increasing *psbA* expression or switching to an alternate variant of the D1 protein allowing reduced O_2_ production and electron flow to and functioning of PS I. Response 4 involves the mechanism from response 3 with SQR providing additional electrons to PS I [10,17,18].

The direct oxidation of sulfide to sulfur by SQR removes the OP inhibiting chemical from the surrounding environment whether or not it provides electrons to fuel AP. Thus, its activity can eventually allow OP. Alternatively, a cyanobacterium that can maintain OP in sulfidic conditions produces O_2_, which can react with the sulfide in its environment through the abiotic reaction:H_2_S + 2O_2_ → SO_4_^2−^ + 2H^+^.(1)

SO_4_^2−^ does not interfere with the OEC and does not affect OP. Therefore, depending on the rate of OP and kinetics of abiotic sulfide oxidation, which may be slow at low temperatures, the production of a small amount of O_2_ might lead to a positive feedback loop by reducing OP inhibition.

**Figure 1 genes-12-00426-f001:**
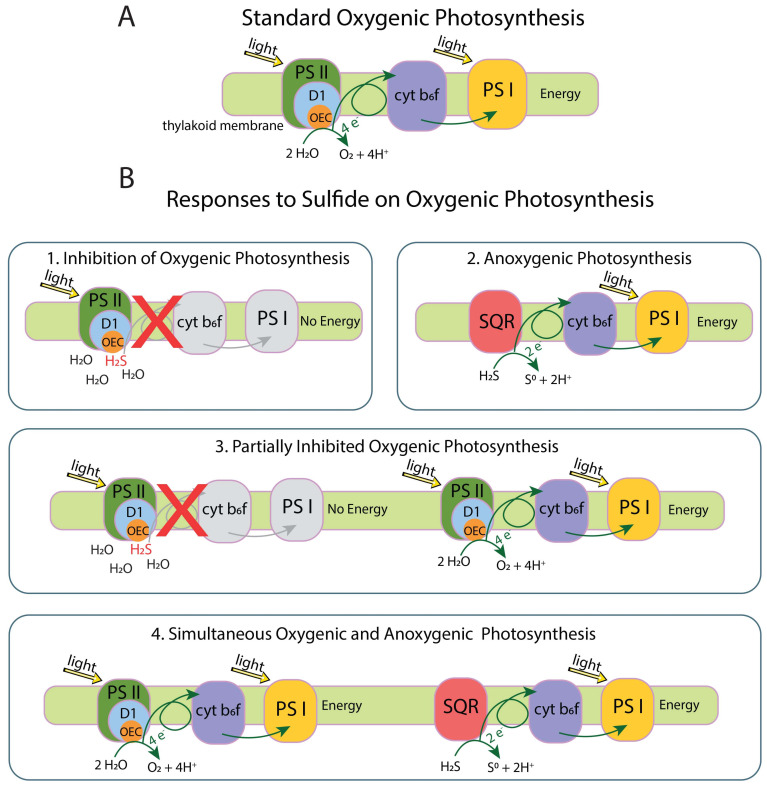
(**A**) standard electron flow in oxygenic photosynthesis. (**B1**) inhibition of oxygenic photosynthesis (OP) by sulfide blocking the oxygen-evolving complex (OEC) in the D1 protein, prohibiting H_2_O from interacting with the OEC and from electrons to flow through the system. (**B2**) anoxygenic photosynthesis (AP) occurs when sulfide quinone reductase (SQR) extracts electrons from sulfide and passes them along the photosynthetic electron transport chain. Sulfide is oxidized to S^0^, which does not interact with the OEC. (**B3**) partially inhibited OP occurs when some D1 proteins are blocked by sulfide while others extract electrons from H_2_O and pass them along to carry out OP. Excess O_2_ produced from OP will oxidize sulfide to sulfate, removing sulfide from the environment. Some cyanobacteria increase *psbA* expression to replace the D1 protein in response to stress, which may support this response. (**B4**) some cyanobacteria can do OP and AP at the same time, or alternate between the two processes, until sulfide is fully depleted. Oxygenic photosynthesis protein complex image modified from the Kyoto Encyclopedia of Genes and Genomes (KEGG) [19,20].

The D1 protein is directly affected by sulfide, and many cyanobacteria have multiple *psbA* genes encoding this protein. One model for reducing sulfide inhibition is altering the production of the D1 protein, which is essential for extracting electrons from H_2_O for OP. The effects of sulfide on D1 protein production have not been studied in cyanobacteria, although the effects of low light and 3-(3,4-dichlorophenyl)-1,1-dimehtylurea (DCMU) have [21,22,23]. DCMU blocks the electron flow from PS II to PS I, which is similar to sulfide due to preventing electron flow to PS I by blocking the extraction of electrons from H_2_O. Understanding how DCMU affects OP may provide insights as to how cyanobacteria deal with sulfide. In response to inhibition from light or DCMU, some cyanobacteria increase their expression of *psbA* to support replacement of the D1 protein [22,23,24]. Cyanobacteria can also respond to some environmental stressors by using different types of D1 proteins. The D1 proteins are divided into four groups: One standard group present in all cyanobacteria that is used under normal conditions, a version specialized for microaerobic conditions, a version for red light, and a nonfunctional version to support nitrogen fixation [25]. To allow flexible responses to environmental conditions, most cyanobacteria contain multiple copies of *psbA*, which allows cyanobacteria to increase expression or use the version of the protein most appropriate for the environmental conditions [22].

Instead of oxidizing sulfide with excess O_2_, some cyanobacteria directly oxidize sulfide to sulfur with SQR [26,27]. Cyanobacteria able to do AP pass these electrons through the quinone pool to PS I [28,29]. Some cyanobacteria can switch between OP and AP, while others can do both simultaneously, with both H_2_O and sulfide donating electrons [16,29]. However, some cyanobacteria, such as *Aphanothece halophytica*, that have *sqr* can survive in the presence of sulfide but not grow, suggesting that the electrons are not shuttled to an energy-producing pathway [26,28].

If *P. pseudopriestleyi* can create an oxygen oasis in the sulfidic benthic environment in Lake Fryxell [12], then it must be able to perform photosynthesis in the presence of sulfide or oxidize the sulfide before producing O_2_. To evaluate the genomic potential for tolerance mechanisms, we obtained a metagenome-assembled genome (MAG) from a natural sample and an enrichment culture of *P. pseudopriestleyi*. We use this MAG to evaluate the genomic potential for survival in elevated sulfide, low light, and cold temperatures and build on the phylogenetic characterization of the *P. pseudopriestleyi* 16S rRNA gene sequence in Jungblut et al. [13]. This study presents an investigation into the metabolic potential of this MAG to gain a better understanding of the connection between metabolic potential and environmental function with implications for primary productivity in sulfidic environments throughout Earth history.

### Site Description

Lake Fryxell is a perennially ice-covered lake located at 77°36′ S 162°6′ E in the McMurdo Dry Valleys of east Antarctica. It is 5 times 1.5 km with a maximum depth of ~20 m [30]. The floor of Lake Fryxell is covered with photosynthetically active microbial mats to depths of ~10 m [12]. At the 9.8 m sampling depth, the lake floor is covered by flat prostrate mats dominated by a single diatom species and *P. pseudopriestleyi*.

The lake receives water from thirteen glacial meltwater streams [31]. Evaporation and ablation from the surface allow for water balance, as no streams flow out from the lake [32]. Salts remaining in the lake water and the historical balance of inflow and sublimation have led to density stratification of the lake water [13,33]. At the 9.8 m sampling depth, the salinity is approximately 4 mS cm^−1^ with 1.2 M NaCl [34,35,36], and sulfide (H_2_S + HS^−^) was present based on diver observations (rotten egg smell). Above 9 m, sulfide was undetectable, and below 10 m, sulfide concentration was 69.9 µM and increased to 1210 µM at the bottom of the lake [30].

Water temperature varies from 2.4 to 2.7 °C, and pH varies from 7.50 to 7.52 along a dive transect established from 8.9 m to 11.0 m in depth [13]. The water column has a sharp oxycline; dissolved oxygen is super saturated below the ice cover to a depth of 9.1 m where it decreases rapidly [33,37]. At 9.8 m depth, there is no O_2_ in the water column, but a microlayer of 50 μmol O_2_ L^−1^ is at least transiently present in the top ~1 mm of the mat [12].

Irradiance is highly seasonal. The lake experiences four months of darkness in the winter, followed by two months of diurnal light variations in spring, four months of continuous summer illumination, and two months of autumn diurnal light variations [31]. Even at peak illumination during the summer, only 0.5–3% of incident light penetrates the ice cover, and light is further attenuated by planktonic communities in the water column [12,38]. A daily average of 1–2 μmol photons m^−2^ s^−1^ reaches the mat at ~10 m water depth [12]. The most penetrating waveband at ~10 m in Lake Fryxell is 520–580 nm [39].

## 2. Materials and Methods

### 2.1. Sulfide Concentrations

Water samples (12 mL) for total sulfide analysis were collected on 17 January 2020 from within 10 mm of the lake floor by a diver using syringes. On return to the surface, samples were immediately injected into glass tubes and preserved for later analysis with 0.2 mL of 2 M zinc acetate. On return to New Zealand, samples were analyzed using the methylene blue method from Standard Methods for the Examination of Water and Wastewater (21st Edition) from the American Public Health Association [40].

### 2.2. Field Work

Samples of the microbial mat were collected in November 2012. Divers accessed the lake through a hole melted in the ice cover and sampled the microbial mat at 9.8 m depth [41]. Sampling and dissection of samples were performed using sterile technique. Briefly, divers used spatulas to cut samples of the mat and transfer them to plastic boxes underwater. In the field lab, mat samples were dissected according to morphology and pigmentation of layers. A blue-green biofilm from 9.8 m water depth was dominated by a single cyanobacterial morphotype based on field microscopy. Samples of this biofilm were peeled off the top of a prostrate microbial mat using sterile forceps [13]. Samples for metagenomic sequencing were preserved in the field within a few hours of collection with Xpedition Soil/Fecal DNA MiniPrep kit (Zymo Research, Irvine, CA, USA) and stored on ice. They were shipped to UC Davis where they were stored at −80 °C until they were processed for sequencing. One subsample for culturing was transferred to a sterile plastic vial filled with lake water that was filtered through a sterile syringe and 0.2 µm syringe filter. The cyanobacteria culture was stored at ambient indoor light at the lakeside laboratory for approximately ten days and then shipped to the Natural History Museum, London, UK where it was grown in BG11 liquid medium at 10 °C, and 24 h light at an average of 9.25 µmol photons m^−2^ s^−1^ [42].

### 2.3. DNA Extraction and Sequencing

DNA from the blue-green biofilm subsample was extracted from frozen samples using an Xpedition Soil/Fecal DNA MiniPrep kit (Zymo Research, Irvine, CA, USA) as per manufacturer instructions. Metagenomic sequencing on mat samples was performed at the Genome Center DNA Technologies Core at the University of California using the Illumina HiSeq 2500, PE250 platform. Illumina’s Nextera DNA Kit was used for library preparation (Oligonucleotide sequences (c) 2007–2013 Illumina, Inc., San Diego, CA, USA) 

DNA was extracted from the cyanobacteria enrichment culture from Lake Fryxell using the MoBio Powerbio DNA extraction kit according to the manufacturer’s instructions. The culture was sequenced on the Illumina HiSeq platform (2000 PE 100, Illumina, Inc., San Diego, CA, USA) at the University of Michigan DNA Sequencing Core.

### 2.4. Bioinformatics Analysis

The sequencing of the biofilm sample from 9.8 m depth resulted in 3,911,904 reads. The biofilm sample data were quality filtered to Q20, and forward and reverse reads were joined using PEAR v0.9.6 [43]. Singletons and replicates with fewer than 10,000 reads were removed from downstream analysis. Sequencing of the culture resulted in 47,243,886 reads. For the lab culture data, trimmomatic v0.36 was used to trim sequencing adaptors with a LEADING and TRAILING parameter of 3, a SLIDINGWINDOW parameter of 4:15, and a MINLEN parameter of 25 [44]. The interleave-reads.py script from khmer v2.1.2 was used to interleave the reads [45]. The biofilm sample and lab-cultured sample were assembled separately and by coassembly with MEGAHIT v1.1.2 [46]. QUAST v4.4 was used to generate assembly statistics [47]. Mapping of both sets of reads to the coassembly was done with bwa v2.3 and samtools v1.9 [48,49]. Anvi’o v2.2.2 was used to bin and visualize the samples with the CONCOCT binning algorithm [50,51]. CheckM v1.0.7 was used to assess the quality of the bins and assign phylogeny based on marker genes of interest [52]. Taxonomy of the bins was assigned using the Genome Taxonomy Database (GTDB-Tk) on KBaseGhostKOALA v2.2 and Prokka v1.11 were used to annotate genes in the cyanobacterial bin of interest [53,54,55,56,57,58,59,60,61,62,63,64]. To refine the bin, spacegraphcats was used to extract additional content of the bin with a k size of 21 [65]. The code used for the analyses presented here is available at https://github.com/jessicalumian/fryxell-phormidium, accessed on 1 March 2021.

A custom Basic Local Alignment Search Tool (BLAST) database containing a reference amino acid sequence was constructed using the makeblastdb command in BLAST+, and D1 protein sequences from the *P. pseudopriestleyi* MAG were retrieved by using a blastx search with an e value of 1e-20 [66]. Subsequent analysis was performed on XSEDE Cipres Science Gateway [67]. D1 protein sequence fragments were aligned to D1 and D2 protein sequences compiled by Cardona et al. [25] with ClustalW v2.1 using standard parameters [68]. A best-fit model of evolution of LG + G4 was selected with ModelTest-ng v0.1.5 using maximum likelihood for the tree topology parameter and the discrete gamma rate categories option was selected for the candidate model’s rate heterogeneity parameter [69]. A phylogenetic tree was generated with RAxML-HPC2 v8.2.12 using a protein gamma model with an LG substitution matrix and 1000 bootstrap iterations [70].

To determine if the SQR in the *P. pseudopriestleyi* is type I or type II, the SQR amino acid sequence from the MAG was aligned to type I and type II references from Shahak and Hauska 2008 [71]. The sequences were aligned with ClustalW v2.1 and then trimmed with TrimAl v1.2.59 using standard parameters [68,72]. A best-fit model of evolution of WAG + G4 was selected with ModelTest-ng v0.1.5 using maximum likelihood for the tree topology parameter and the discrete gamma rate categories option was selected for the candidate model’s rate heterogeneity parameter [69]. A phylogenetic tree was generated with RAxML-HPC2 v8.2.12 using a protein gamma model, WAG substitution matrix and 1000 bootstrap iterations [70].

The average nucleotide identity (ANI) was calculated between the *P. pseudopriestleyi* MAG and available Antarctic cyanobacteria genomes (*Leptolyngbya* sp. BC1307, accession number NRTA00000000.1, *Aurora vandensis*, accession number JAAXLU010000000, *Synechococcus* sp. CS-601, accession number CP018091, and *Phormidesmis priestleyi* ULC007, accession number MPPI01000000) [73,74,75,76]. The ANI was also calculated between the MAG and an Arctic cyanobacterium closely related to an Antarctic strain (*Phormidesmis priestleyi* BC1401, accession number LXYR01000000) [77], and the closest related genome according to 16S rRNA gene sequence (*Oscillatoria acuminata* PCC 6304, accession number CP003607.1) [78]. Calculations were performed using the ANI calculator from the Kostas lab using the default parameters [79]. The alignment options required a 700 bp minimum length, 70% minimum identity, and 50 minimum alignments. The fragment option window size was set to 1000 bp with a step size of 200 bp.

To identify the presence *P. pseudopriestleyi* in other locations, the 16S rRNA gene sequence reported in Jungblut et al. [13] (accession number KT347094) was aligned with blastn to sequences from Jungblut et al. [80] (accession numbers AY541534 and AY541575) and Taton et al. [81] (accession number DQ181670). The phylogenetic tree of the 16S rRNA gene sequence of *P. pseudopriestleyi* and the most closely related operational taxonomic units (OTUs) in Jungblut et al. [13] was used for context.

## 3. Results

### 3.1. Sulfide Concentrations

Total sulfide ([H_2_S] + [HS^−^]) was measured to be <0.01 mg L^−1^ at 9.5 m and above. At 9.8 m, total sulfide was 0.091 mg L^−1^ and rose to 2.2 mg L^−1^ at 10.7 m depth. See Table 1 for all measurements.

### 3.2. Assembly and Binning Statistics

Thirty-one bins were retrieved from the coassembly from the environmental biofilm samples and the strain *P. pseudopriestleyi* (see Table 2 for assembly statistics and Table 3 binning statistics), but there was only one bin belonging to the phylum Cyanobacteria based on comparison with GTDB-tk on KBASE database [13]. This bin is 5.97 MB with 42,524 reads, 678 contigs, 4738 protein coding genes, and a GC content of 62.32%. The longest contig generated was 400,416 bp. The closest relative to *P. pseudopriestleyi* according to 16S rRNA OTUs, *Oscillatoria acuminata* PCC 6304 has a genome size of 7.7 MB and contains 5,687 protein coding genes. The *P. pseudopriestleyi* MAG is 91.73% complete, has 1.35% contamination and 8.33% strain heterogeneity. Because the bin used for analysis is incomplete, it may not contain genes that are present in the organism’s genome. However, the MAG is sufficiently complete that it can be used to analyze the genomic potential for key metabolisms. A complete list of KEGG annotated genes is available in Appendix A.

### 3.3. Taxonomic Assignment of MAG

The MAG obtained from *P. pseudopriestleyi* was classified as family Oscillatoriaceae and genus *Oscillatoria* based on comparison with GTDB-tk on KBASE database [13]. The *P. pseudopriestleyi* MAG and *O. acuminata* PCC 6304 had an ANI of 88.99%, with a standard deviation of 3.25% based on 14,129 fragments. The *P. pseudopriestleyi* MAG and *P. priestleyi* BC1401 had an ANI of 75.69% with a standard deviation of 5.94% based on 75 fragments. The ANI score between the *P. pseudopriestleyi* MAG and *P. priestleyi* ULC007 was 72.03% with a standard deviation of 3.65% based on 72 fragments. Each of the ANIs between the *P. pseudopriestleyi* MAG and *A. vandensis, Synechococcus* sp. *CS-601*, and *Leptolyngbya* sp. BC1307 were less than 70%. These results suggest that the *P. pseudopriestleyi* MAG was most similar to *O. acuminata* [57,79,82]. However, additional cyanobacteria strains in Oscillatoriaceae need to be isolated and sequenced from Antarctica to allow phylogenomic interference to better resolve the relationship between *Phormidium* and *Oscillatoria.*

### 3.4. Photosynthetic and Electron Transport Machinery

The *P. pseudopriestleyi* MAG contains genes for the pigments phycoerythrocyanin, phycocyanin, and allophycocyanin, which have absorption peaks at 575, 620, and 650 nm, respectively, and lacks the gene for phycoerythrin, the light-harvesting protein that absorbs wavelengths from 495 to 560 nm [83] (Table 4). The genes for the D1/D2 protein cluster in PS II (*psbA* and *psbD*) are present along with two genes for proteins that hold the Mn_4_CaO_5_ OEC cluster in place (*psbP* and *psbO*) [25]. The MAG contains four of the eight subunits that make up the cytochrome b_6_f complex (*petB*, *petD*, *petA*, and *petC*). Notably, *petA* codes for apocytochrome f and *petC* codes for the iron-sulfur subunit within the b_6_f complex. The genes for the remaining subunits, *petL*, *petM*, *petN*, and *petG*, were not identified in the bin (Table 4). The majority of PS I genes are present in the MAG, including the core chlorophyll dimer made up of *psaA* and *psaB* (Table 4). Genes for both ferredoxin (*petF*) and ferredoxin-NADP+ reductase (*petH*) are present. The MAG does not contain *petE*, which codes for the electron transport protein plastocyanin that connects PS II to PS I, however cytochrome c6 encoded by *petJ* can perform the same function [84]. The MAG has all the genes necessary to create an F-type ATPase. Notably, the MAG has a type II *sqr* (Figure 2), which is necessary for sulfide oxidation in AP. The MAG also contains genes encoding arsenic resistance (*arsR* and *acr3*) that may be involved with transcriptional regulation of *sqr* [85].

The MAG contains all genes for a type 1 NADH dehydrogenase except for *ndhN*, subunit n of the complex. The succinate dehydrogenase gene *sdhD,* encoding a membrane anchor subunit, is absent but *sdhA, shdB,* and *sdhC* are present. The genes *ctaC, ctaD,* and *ctaE* for aa_3-_type cytochrome c oxidase genes are present but *ctaF* is not. The genes for cytochrome bd-quinol oxidase *cydA* and *cydB* are present, but *cydX* is not. Genes for alternative respiratory terminal oxidase and plastid terminal oxidase were not found in the MAG [86,87].

Because of the importance of the D1 protein to OP, we examined *psbA* sequence in more detail. Although spacegraphcats software was used to refine the MAG, a full copy of the gene for the D1 protein could not be obtained. Fragments of the D1 protein sequence are present, including the C-terminal and N-terminal portions ranging between 80 and 264 amino acids out of the full length 360 amino acids protein sequence (Alignment S1). Additionally present are all seven of the amino acids involved in ligating with the Mn_4_CaO_5_ cluster in the OEC: Asp170, Glu189, His332, Glu333, His337, Asp342, and Ala344 [88,89]. Each of these amino acids were present in fragments that contained the appropriate part of the protein sequence for psbA. A phylogenetic tree was constructed of the MAG’s D1 fragments and reference sequences from all four D1 groups, and all fragments grouped closely with group 4 D1 proteins, demonstrating that the MAG does not contain an alternative version of the D1 protein (Figure 3). Where overlapping, the sequences of the fragments are not identical, indicating the presence of at least two copies of the D1 gene in the MAG.

### 3.5. Metabolic Pathways

The MAG contains genetic capacity for the Calvin cycle for carbon fixation. The MAG also contains genes for glycolysis via the Embden–Meyerhof pathway except for *tpiA* encoding for triosephosphate isomerase. Capacity for the tricarboxylic acid cycle is present except for genes for 2-oxoglutarate dehydrogenase, which is absent in cyanobacteria, and fumarate hydratase. The MAG contains genes for the pentose phosphate pathway and glycogen and trehalose biosynthesis pathways. It also contains full capacity for the initiation, elongation, and β-oxidation of fatty acids (Table 5).

All genes necessary for assimilatory sulfate reduction are present in the MAG (*sat, cysNC, cysH, and sir*), but essential genes for dissimilatory sulfur metabolism are not (*aprAB* and *dsrAB*). Additionally, the MAG has the genes necessary for a sulfate ion transport system through a membrane (*cysPUWA*, *sbp*). The presence of *narB* and *nirA* indicate capacity for assimilatory nitrate reduction, but the MAG does not contain genes for dissimilatory nitrogen metabolism or nitrogen fixation [90]. Although 19 genes associated with methane metabolism were found in the MAG, most of them are involved with various biosynthesis pathways, and there is not a full pathway for methanogenesis or methanotrophy. Notably, *hdrA2, hdrB2,* and *hdrC2*, are present, which code for heterodisulfide reductase, an enzyme typically found in methanogens. Previous work has found these genes at 9.8 m depth in Lake Fryxell, and consistent with the content of the MAG, the capacity for methanogenesis (*hdrD*) was absent [91]. The presence of *hdrB* may indicate capacity for flavin-based electron bifurcation [92].

### 3.6. Genes Implicated in the Adaption to Environmental Stress

*P. pseudopriestleyi* encodes some genes related to osmotic stress. Specifically, the MAG contains several genes related to sodium and potassium antiporters *(nhaS2*, *nhaS3*, *mrpA*, *mrpC*, *trk*, and *ktr*). Additionally, the MAG contains *treZ* and *treY*, which support a trehalose biosynthesis pathway, a compatible solute that has been found to have membrane protective features, particularly in filamentous, mat-forming cyanobacteria strains [93]. Trehalose has also been found in cyanobacteria tolerant of desiccation [94,95,96]. The MAG does not contain genes related to glucosylglycercol or glycine betaine, which are compatible solutes that have been identified in halotolerant and halophilic cyanobacteria strains [97]. Sucrose can also be used as an osmolyte in cyanobacteria, and the MAG contains *spsA*, encoding sucrose phosphate synthase which supports the production of sucrose 6-phosphate, but not *spp*, encoding sucrose phosphate phosphatase, which is necessary to produce sucrose [98].

Some cyanobacteria synthesize the compounds scytonemin and mycosporine to overcome the harmful effects of long-term UV radiation exposure [99]. The MAG contains no genes related to the biosynthesis of these compounds, though the pathways are not fully understood [100,101]. The photoprotective proteins such as orange carotenoid protein (*ocp*) and fluorescence recovery protein (*frp*) protect against high light stress by converting excess excitation energy to heat [87], but the MAG does not contain either of these genes. Another method of dealing with high light stress is to divert electrons away from the photosynthetic electron transport chain using electron valves to prevent over-reduction of photosynthetic machinery. The MAG contains genetic capacity for cyanobacterial electron valves flavodiiron proteins Flv1-4 and another cyanobacterial bidirectional hydrogenase. Additionally, the MAG contains genes for terminal oxidases cytochrome *bd*-1 and cytochrome c oxidase, which can act as electron valves [87]. Carotenoids are another important molecule for photooxidative stress. The MAG contains *crtE, crtB, crtP, crtH, crtQ, cruF, cruG, crtR,* and *crtO*, allowing for carotenoid biosynthesis of myxol-2′-dimehtylfucoside, β-carotene, zeaxanthin, echinenone, and 3’-hydroxyechinenone. It is missing *cruE* and *cruH*, and thus does not demonstrate a capacity to produce the carotenoid synechoxanthin [102,103,104]. Additional carotenoid biosynthesis genes *crtI_1, crtI_2,* and *crtI_3* for lycopene and neurosporene phytoene desaturase are present. Although chlorophyll F has been shown to support near-infrared OP, the gene for chlorophyll F synthase (*chlf*) is not present in the MAG [105].

The MAG contains genes relating to replication, transcription, and translation associated with cold-tolerant organisms. It has the gene for DNA gyrase (*gyrA*) that helps uncoil DNA that is tightly wound at cold temperatures [106]. The MAG contains genes associated with cold-adapted ribosomal function, including ribosome-binding factor A (*rbfA*) as well as genes for translational factors Initiation Factor 1 and 2 (IF1 and IF2) (*infA*, *infB*) [106]. Genes for a ribosomal rescue system *ssrA* and *smpB* are also present. Genes for delta (12)-fatty-acid desaturase (*desA*) and NADPH-dependent stearoyl-CoA 9-desaturase (*desA3*) are present in the MAG. These produce unsaturated and branched fatty acids, which help organisms maintain membrane integrity at lower temperatures. None of the *csp* cold shock proteins are present in the MAG.

Cyanophycin is a copolymer of aspartic acid and arginine that stores nitrogen for when environmental nitrogen levels become deficient [107]. The MAG contains genes for both cyanophycin synthetase to build the polymer (*cphA*) and cyanophycinase (*cphB*) to break it down. 

## 4. Discussion

### 4.1. Sulfide Resistance in P. pseudopriestleyi

Knowing the concentration of sulfide present in the mat at 9.8 m depth is important for understanding the extent of OP inhibition experienced by *P. pseudopriestleyi* in Lake Fryxell during the winter to spring transition. The early spring concentration was likely higher than the 0.091 mg L^−1^ sulfide measured at this depth (Table 1), because water samples were collected in mid-January, several months after initiation of oxygenic photosynthesis, which results in sulfide oxidation. Thus, the sulfide concentrations reported here represent minimums for early spring OP inhibition. 

Both prior research [12,13] and our field work demonstrate that *P. pseudopriestleyi* sustains OP in an environment with at least 0.091 mg L^−1^ sulfide which suggests the presence of a tolerance mechanism for sulfide. One option for a sulfide tolerance mechanism is that the D1 protein is repaired and replaced while OP occurs (response 3 or 4 in Figure 1). Another possible mechanism is that SQR production supports AP in the presence of sulfide (response 2 or 4 in Figure 1). Additionally, the low irradiance in Lake Fryxell may contribute to *P. pseudopriestleyi*’s sulfide tolerance by minimizing photodamage to the D1 protein and allowing OP to occur at low rates (response 2 in Figure 1). 

Sulfide inhibits water from interacting with the water-splitting Mn_4_CaO_5_ cluster of the OEC in the D1 protein of PS II (encoded by *psbA*), preventing the use of H_2_O as an electron donor and consequently OP [15]. Expression of *psbA* is increased in cyanobacteria exposed to light stress or DCMU, which inhibits the quinone binding site of PS II and thus electron transfer to PS I [21,22,23]. Similar increased expression of *psbA* may also occur in response to sulfide stress, although this effect has not been studied in cyanobacteria. *P. pseudopriestleyi* has multiple copies of the D1 protein, which may assist with increasing expression of *psbA*. Even though *P. pseudopriestleyi* grows in a low O_2_ environment, the MAG appears to only contain genetic capacity for the standard group 4 D1 protein, and there is no evidence that the organism has a microaerobic D1 protein. Thus, although the MAG is incomplete, *P. pseudopriestleyi* does not appear to use an alternative D1 as part of its sulfide tolerance strategy. If its tolerance mechanism is related to the D1 protein, *P. pseudopriestleyi* likely overcomes sulfide inhibition by either increasing *psbA* expression (response 2 in Figure 1) or activating a mechanism that has not been identified. In this scenario, O_2_ produced from OP will oxidize sulfide in the abiotic reaction presented in reaction 1 at a quick enough rate to prevent sulfide inhibition. This oxidation is passive and may be kinetically very slow at Lake Fryxell temperatures (2 °C), but includes positive feedback between decreasing sulfide and increasing capacity for O_2_ production. 

Alternatively, *P. pseudopriestleyi* may employ AP by directly oxidizing sulfide to S^0^ with the membrane protein SQR (response 3 or 4 in Figure 1). This process would deplete sulfide, allowing OP. Cyanobacteria capable of AP pass electrons from SQR oxidation through the quinone pool to PS I to harvest energy [16]. If *P. pseudopriestleyi* transfers electrons from SQR oxidation to PS I, it gains energy through AP, while sulfide is being depleted. The MAG contains the *sqr* gene and may be capable of this response. However, laboratory incubation experiments are necessary to determine if the cyanobacterium performs AP, regardless of *sqr* in the genome [10,26,108].

### 4.2. Ecology of P. pseudopriestleyi in Lake Fryxell

The seasonality of Lake Fryxell controls how *P. pseudopriestleyi* shapes the redox potential of its environment by producing O_2_ through OP. Complete darkness from mid-April to mid-August allows sulfide to accumulate in the mats, creating a reduced, sulfidic environment [12]. As light becomes available starting in mid-August, OP or AP may initiate, leading to the oxidation of sulfide. Constant summer irradiance starts at the end of October, with irradiance sufficient to allow OP, resulting in the accumulation of O_2_ the mats [12]. OP slows down as light levels fall from mid-February to mid-April and then ceases in total darkness, and sulfide reaccumulates.

The low irradiance in *P. pseudopriestleyi*’s spring and summer environment may contribute to its sulfide resistance. Even during peak irradiance in the summer, the daily mean photon flux at 9.8 m depth is 1–2 µmol m^−2^ s^−1^. Previous research demonstrates that low irradiance allowed the hot spring cyanobacteria *Planktothrix* str. FS34 to perform OP uninhibited in up to 230 µM sulfide (or 7.83 mg L^−1^ sulfide) even though sulfide inhibited photosynthesis at higher light fluxes [18]. Lower light levels may reduce photo-damage on the photosensitive D1 protein, allowing more D1 proteins in the thylakoid membrane to perform sulfide-tolerant OP. If sulfide levels are below the threshold for OP inhibition, the O_2_ from OP will oxidize sulfide, eventually allowing O_2_ to accumulate. If this mechanism is happening in Lake Fryxell, the balance of irradiance and sulfide levels in Lake Fryxell plays an important role in *P. pseudopriestleyi*’s ability to perform OP in an extreme environment.

The effect of low irradiance on *P. pseudopriestleyi* is amplified by a partial mismatch between the wavelengths of light available and the pigments it can produce. The most penetrating waveband at 9 m in Lake Fryxell is 520–580 nm because the ice cover transmits blue light and absorbs wavelengths longer than 600 nm [38,109]. The MAG contains genes for phycoerythrocyanin, phycocyanin, and allophycocyanin, which have peak absorptions at 575, 620, and 650 nm, respectively. Without the genes for phycoerythrin, absorption of wavelengths between 495–560 nm is limited. The combination of pigments and available light suggests that phycoerythrocyanin harvests most of the light for photosynthesis in the Lake Fryxell environment. Although there is a mismatch between available light and pigments for photosynthesis, the low irradiance of the environment is consistent with the absence of UV exposure genes in the MAG.

Continuous illumination during summer, even at low levels, requires a persistent nutrient source. The MAG has genes to create and break down cyanophycin, a nitrogen storage molecule synthesized by cyanobacteria. This may aid *P. pseudopriestleyi* in meeting peak nitrogen demands during the summer in Lake Fryxell, which is nitrogen limited [110].

### 4.3. P. pseudopriestleyi in Other Environments

In addition to dominating a narrow sulfide-rich photic benthic zone of Lake Fryxell, *P. pseudopriestleyi* has been reported from several ponds and lakes across Antarctica, based on 16S rRNA gene analyses [80,81]. *P. pseudopriestleyi*’s 16S gene sequence reported in Jungblut et al. [13] has a 99.45%, 91.28%, and 99.9% identity to 16S rRNA gene sequences from Antarctic Salt Pond, Fresh Pond, and Ace Lake, respectively (accession numbers AY541534, AY541575, and DQ181670) according to a blastn search. Salt and Fresh Ponds are meltwater ponds on the McMurdo Ice Shelf that experience high UV radiation [111]. Salt Pond is hypersaline with a conductivity of up to 52.9 mS cm^−1^, with high salinity originating from diluted seawater or sulfate salts from chemical weathering of sedimentary material [80,112,113]. Ace Lake is considered hyposaline with conductivity of 25.4–26.4 mS cm^−1^ and is permanently stratified [81,114]. *P. pseudopriestleyi* is present in both locations, suggesting it is adapted to varying levels of salinity, UV, and high light stress in addition to sulfide. Future DNA sequencing beyond 16S amplicons in these environments may reveal whether or not *P. pseudopriestleyi* possesses additional stress tolerance genes related to these conditions that are absent in the Lake Fryxell MAG.

## 5. Conclusions

*P. pseudopriestleyi* is the first example of a cyanobacterium capable of sulfide-tolerant OP in a cold, Antarctic environment. The MAG reconstructed for *P. pseudopriestleyi* revealed a genome that is consistent with its ability to produce O_2_ in a sulfidic, cold, low-light environment of the perennially ice-covered Lake Fryxell, Antarctica. The MAG has a genomic capacity to deal with sulfide with multiple copies of a *psbA*, the D1 protein that is the site of water splitting, or by using SQR to deplete sulfide through AP or through sulfide oxidation. The low light levels at 9.8 m in Lake Fryxell may also contribute to its sulfide tolerance. Thus, there are likely several methods for dealing with sulfide stress on OP in a low light environment. Sulfide tolerance varies widely among cyanobacteria, and the consideration of light level may have implications on a response to sulfide and should be studied further. Specifically, microelectrode measurements combined with gene expression data are likely to uncover the molecular mechanism *P. pseudopriestleyi* uses to perform sulfide-tolerant OP.

Besides Lake Fryxell, *P. pseudopriestleyi* has been found in shallow freshwater and hypersaline ice shelf melt water ponds and lakes, indicating a widespread distribution in Antarctica, and ability to thrive in a range of environmental conditions. If genomic data from *P. pseudopriestleyi* living in these environments can be obtained, a comparison with the MAG from Lake Fryxell can provide insight about the effects of various Antarctic conditions, such as UV exposure, high light levels, or salinity, on a genome. Additionally, the isolation and sequencing of other Antarctic cyanobacteria will allow for more in-depth genomic comparisons between Antarctic cyanobacteria genomes beyond the three genomes currently published.

A deeper understanding of the ecology of cold cyanobacteria ecosystems will provide insights into the production of O_2_ through Earth history. Specifically, primary productivity during “Snowball Earth” glaciations was required to sustain the biosphere through these climatic crises. In some cases, sulfide appears to have been abundant [3,4,5,115], suggesting that *P. pseudopriestleyi* may provide a model for how OP persisted in some “Snowball Earth” ecosystems.

## Figures and Tables

**Figure 2 genes-12-00426-f002:**
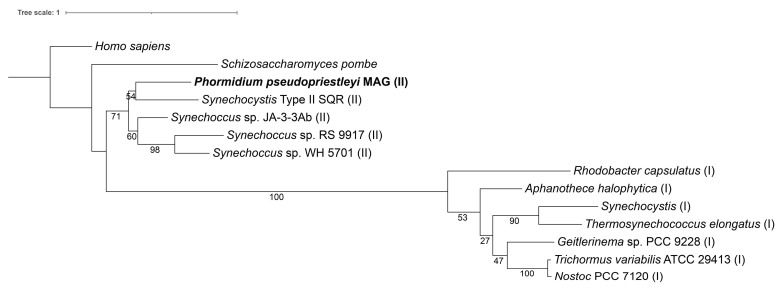
A maximum likelihood tree of type I and II SQR amino acid sequences from Shahak and Hauska, 2008. Homo sapiens (accession number AAH16836), *Schizosaccharomyces pombe* (accession number CAA21882), *Phormidium pseudopriestleyi* (presented in paper), *Synechocystis* SQR-type II (accession number WP_010872226), *Synechococcus* strain sp. JA-3-3Ab (accession number ABD00861), *Synechococcus* sp. RS 9917 (accession number EAQ69368), *Synechococcus* strain WH 5701 (accession number EAQ74835), *Rhodobacter capsulatus* (accession number CAA66112), *Aphanothece halophytica* (accession number AAF72963), *Synechocystis* SQR-type I (accession number WP_011153573), *Thermosynechococcus elongatus* (accession number WP_011056143), *Geitlerinema* sp. PCC 9228 (formerly known as *Oscillatoria limnetica*, accession number AAF72962), *Trichormus variabilis* ATCC 29413 (formerly known as *Anabaena variabilis* accession number ABA22985), and *Nostoc* PCC 7120 (accession number WP_010998645). A type I SQR is indicated by (I) after the organism name, while SQR type II is indicated by (II).

**Figure 3 genes-12-00426-f003:**
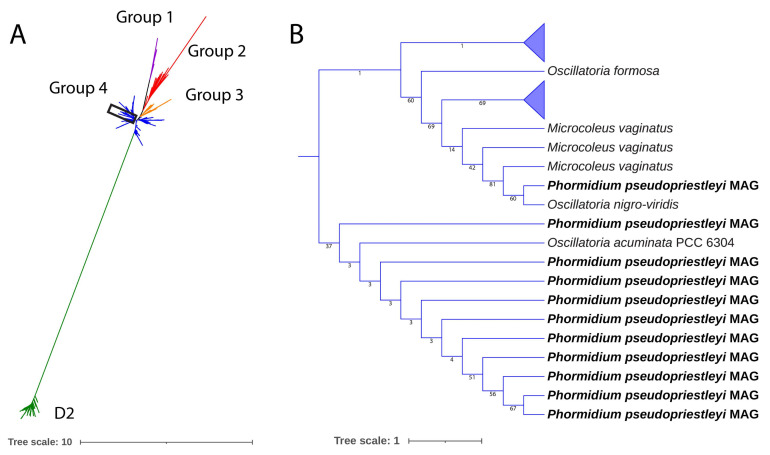
(**A**) a maximum likelihood tree of D1 and D2 proteins of sequences presented in Cardona et al. [25] and the 11 D1 protein fragments in the *P. pseudopriestleyi* MAG. All *P. pseudopriestleyi* fragments (enclosed in the rectangle) grouped with group 4 D1 proteins. (**B**) region of the tree showing *P. pseudopriestleyi* and the most closely related D1 proteins.

**Table 1 genes-12-00426-t001:** Total sulfide measurements on Lake Fryxell water samples collected on 17 January 2020.

Depth (m)	Total Sulfide (mg L^−1^)
8	<0.01
9	<0.01
9.5	<0.01
9.8	0.091
10.1	0.571
10.3	0.885
10.7	2.238

**Table 2 genes-12-00426-t002:** Quality metrics of coassembly generated from sequencing of the mat and laboratory culture samples. All statistics are from QUAST v4.4, except for mapping statistics which were generated from samtools v1.9 using the flagstat parameter.

Metric	Coassembly
Total number of contigs	137,226
Longest contig length	453,833
Total length (bp)	175,382,959
GC content (%)	61.61
N50	1441
Reads mapped from culture (%)	79.80
Reads mapped from mat (%)	54.27
Total reads mapped (%)	77.85
Number of contigs ≥ 0 bp	336,522
Number of contigs ≥ 1000 bp	41,881
Number of contigs ≥ 5000 bp	2359
Number of contigs ≥ 10.000 bp	814
Number of contigs ≥ 25.000 bp	303
Number of contigs ≥ 50.000 bp	152

**Table 3 genes-12-00426-t003:** Quality metrics of the *P. pseudopriestleyi* metagenome-assembled genome (MAG). All statistics are from QUAST v4.4, except for completion and contamination statistics which were generated from CheckM v1.0.7 and the number of protein coding genes from Prokka v1.11.

Metric	MAG
Total number of contigs	678
Longest contig length	44,245
Total length (bp)	5,965,908
GC content (%)	47.43
N50	10,908
Completion (%)	91.73
Contamination (%)	1.35
Number of protein coding genes	4738
Number of contigs ≥ 0 bp	678
Number of contigs ≥ 1000 bp	678
Number of contigs ≥ 5000 bp	458
Number of contigs ≥ 10.000 bp	203
Number of contigs ≥ 25.000 bp	20
Number of contigs ≥ 50.000 bp	0

**Table 4 genes-12-00426-t004:** Phycobilisome, photosynthesis, and respiratory machinery genes present in the *P. pseudopriestleyi* MAG generated with GhostKoala v2.2.

Complex	Gene	Presence	Function
Allophycocyanin	*apcA*	Yes	Allophycocyanin α subunit
	*apcB*	Yes	Allophycocyanin β subunit
	*apcC*	Yes	Phycobilisome core linker protein
	*apcD*	Yes	Allophycocyanin-B
	*apcE*	Yes	Phycobilisome core-membrne linker protein
	*apcF*	Yes	Phycobilisome core component
Phycocyanin/ Phycoerythrocyanin	*cpcA*	Yes	Phycocyanin α chain
	*cpcB*	Yes	Phycocyanin β chain
	*cpcC*	Yes	Phycocyanin-associated rod linker protein
	*cpcD*	No	Phycocyanin-associated, rod
	*cpcE*	Yes	Phycocyanobilin lyase α subunit
	*cpcF*	Yes	Phycocyanobilin lyase β subunit
	*cpcG*	Yes	Phycobilisome rod-core linker protein
Phycoerythrin	*cpeA*	No	Phycoerythrin α chain
	*cpeB*	No	Phycoerythrin β chain
	*cpeC*	No	Phycoerythrin-associated linker protein
	*cpeD*	No	Phycoerythrin-associated linker protein
	*cpeE*	No	Phycoerythrin-associated linker protein
	*cpeR*	No	Phycoerythrin-associated linker protein
	*cpeS*	No	Phycoerythrin-associated linker protein
	*cpeT*	No	CpeT protein
	*cpeU*	No	Billin biosynthesis protein
	*cpeY*	No	Billin biosynthesis protein
	*cpeZ*	No	Billin biosynthesis protein
Photosystem II	*psbA*	Yes	Photosystem II P680 reaction center D1 protein
	*psbD*	Yes	Photosystem II P680 reaction center D2 protein
	*psbC*	Yes	Photosystem II CP43 chlorophyll apoprotein
	*psbB*	Yes	Photosystem II CP47 chlorophyll apoprotein
	*psbE*	Yes	Photosystem II cytochrome b559 subunit α
	*psbF*	Yes	Photosystem II cytochrome b559 subunit β
	*psbL*	Yes	Photosystem II PsbL protein
	*psbJ*	Yes	Photosystem II PsbJ protein
	*psbK*	Yes	Photosystem II PsbK protein
	*pskM*	Yes	Photosystem II PsbM protein
	*psbH*	Yes	Photosystem II PsbH protein
	*psbI*	Yes	Photosystem II PsbI protein
	*psbO*	Yes	Photosystem II oxygen-evolving enhancer protein 1
	*psbP*	Yes	Photosystem II oxygen-evolving enhancer protein 2
	*psbQ*	No	Photosystem II oxygen-evolving enhancer protein 3
	*psbR*	No	Photosystem II 10 kDa protein
	*psbS*	No	Photosystem II 22kDa protein
	*psbT*	Yes	Photosystem II PsbT protein
	*psbU*	Yes	Photosystem II PsbU protein
	*psbV*	Yes	Photosystem II cytochrome c550
	*psbW*	No	Photosystem II PsbW protein
	*psbX*	Yes	Photosystem II PsbX protein
	*psbY*	Yes	Photosystem II PsbY protein
	*psbZ*	Yes	Photosystem II PsbZ protein
	*Psb27*	Yes	Photosystem II Psb27 protein
	*psb28*	Yes	Photosystem II 13kDa protein
	*psb28-2*	No	Photosystem II Psb28-2 protein
Photosystem I	*psaA*	Yes	Photosystem I P700 chlorophyll a apoprotein A1
	*psaB*	Yes	Photosystem I P700 chlorophyll a apoprotein A2
	*psaC*	Yes	Photosystem I subunit VII
	*psaD*	Yes	Photosystem I subunit II
	*psaE*	Yes	Photosystem I subunit IV
	*psaF*	Yes	Photosystem I subunit III
	*psaG*	No	Photosystem I subunit V
	*psaH*	No	Photosystem I subunit VI
	*psaI*	Yes	Photosystem I subunit VIII
	*psaJ*	No	Photosystem I subunit IX
	*psaK*	Yes	Photosystem I subunit X
	*psaL*	Yes	Photosystem I subunit XI
	*psaM*	Yes	Photosystem I subunit XII
	*psaN*	No	Photosystem I subunit PsaN
	*psaO*	No	Photosystem I subunit PsaO
	*psaX*	No	Photosystem I 4.8kDa protein
Cytochrome b_6_f Complex	*petB*	Yes	cytochrome b_6_
	*petD*	Yes	cytochrome b_6_f complex subunit 4
	*petA*	Yes	apocytochrome f
	*petC*	Yes	cytochrome b_6_f complex iron-sulfur subunit
	*petL*	No	cytochrome b_6_f complex subunit 6
	*petM*	No	cytochrome b_6_f subunit 7
	*petN*	No	cytochrome b_6_f complex subunit 8
	*petG*	No	cytochrome b_6_f complex subunit 5
Photosynthetic Electron Transport Chain	*petE*	No	plastocyanin
	*petF*	Yes	ferredoxin
	*petH*	Yes	ferredoxin-NADP^+^ reductase
	*petJ*	Yes	cytochrome c6
F-type ATPase	*atpD*	Yes	H^+^/Na^+^ transporting ATPase subunit β
	*atpA*	Yes	F-type H^+^/Na^+^ transporting ATPase subunit α
	*atpG*	Yes	H^+^ transporting ATPase subunit γ
	*atpH*	Yes	F-type H^+^ transporting ATPase subunit δ
	*atpC*	Yes	F-type H^+^ transporting ATPase subunit ε
	*atpE*	Yes	F-type H^+^ transporting ATPase subunit c
	*atpB*	Yes	F-type H^+^ transporting ATPase subunit a
	*atpF*	Yes	F-type H^+^ transporting ATPase subunit b
NADH Dehydrogenase	*ndhC*	Yes	NADH-quinone oxidoreductase subunit 3
	*ndhK*	Yes	NADH-quinone oxidoreductase subunit K
	*ndhJ*	Yes	NADH-quinone oxidoreductase subunit J
	*ndhH*	Yes	NADH-quinone oxidoreductase subunit H
	*ndhA*	Yes	NADH-quinone oxidoreductase subunit 1
	*ndhI*	Yes	NADH-quinone oxidoreductase subunit I
	*ndhG*	Yes	NADH-quinone oxidoreductase subunit 6
	*ndhE*	Yes	NADH-quinone oxidoreductase subunit 4L
	*ndhF*	Yes	NADH-quinone oxidoreductase subunit 5
	*ndhD*	Yes	NADH-quinone oxidoreductase subunit 4
	*ndhB*	Yes	NADH-quinone oxidoreductase subunit 2
	*ndhL*	Yes	NADH-quinone oxidoreductase subunit L
	*ndhM*	Yes	NADH-quinone oxidoreductase subunit M
	*ndhN*	No	NADH-quinone oxidoreductase subunit N
	*hoxE*	Yes	bidirectional [NiFe] hydroganse diaphorase subunit
	*hoxF*	Yes	bidirectional [NiFe] hydroganse diaphorase subunit
	*hoxU*	Yes	bidirectional [NiFe] hydroganse diaphorase subunit
Succinate Dehydrogenase	*sdhC*	Yes	H^+^/Na^+^ transporting ATPase subunit β
	*sdhD*	No	F-type H^+^/Na^+^ transporting ATPase subunit α
	*sdhA*	Yes	H^+^ transporting ATPase subunit γ
	*sdhB*	Yes	F-type H^+^ transporting ATPase subunit δ
Cytochrome c oxidase	*ctaC*	Yes	cytochrome c oxidase subunit 2
	*ctaD*	Yes	cytochrome c oxidase subunit 1
	*ctaE*	Yes	cytochrome c oxidase subunit 3
	*ctaF*	No	cytochrome c oxidase subunit 4
Cytochrome bd complex	*cydA*	Yes	cytochrome bd ubiquinol oxidase subunit I
	*cydB*	Yes	cytochrome bd ubiquinol oxidase subunit II
	*cydX*	No	cytochrome bd ubiquinol oxidase subunit X

**Table 5 genes-12-00426-t005:** Number of genes present in the *P. pseudopriestleyi* MAG in functional categories based on KEGG annotations. A full list of these genes is available in Appendix A.

Category	Complex or System	Number of Genes in MAG	Total Number of Genes in KEGG Category
Phycobilisome Antenna Proteins	Allophycocyanin	6	6
	Phycocyanin/Phycoerythrin	6	7
	Phycoerythrin	0	11
Photosynthesis Machinery	Photosystem II	22	27
	Photosystem I	10	16
	Cytochrome b_6_f complex	4	8
	Photosynthetic electron transport	3	4
	F-type ATPase	8	8
Nitrogen Metabolism	Dissimilatory Nitrate Reduction	0	4
	Assimilatory Nitrate Reduction	2	5
	All Nitrogen Metabolism	9	35
Sulfur Metabolism	Assimilatory Sulfate Reduction	4	7
	Dissimilatory Sulfate Reduction and Oxidation	1	3
	All Sulfur Metabolism	10	54
Carbon Fixation	Carbon Fixation in Photosynthetic Organisms	9	23
Methane Metabolism	Methane Metabolism	19	79

## Data Availability

The high-throughput sequencing data are available via NCBI’s Sequence Read Archive. Metagenomic reads from the Lake Fryxell microbial mats are available at the accessions SAMN09937182-SAMN09937191. The *P. pseudopriestleyi* laboratory culture reads are available at the accession SAMN17478355. The *P. pseudopriestleyi* MAG has been deposited at GenBank under the accession JAFLQW000000000. The version described in this paper is version JAFLQW010000000.

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
