# Peer review of "Metabolic Capacity of the Antarctic Cyanobacterium Phormidium pseudopriestleyi That Sustains Oxygenic Photosynthesis in the Presence of Hydrogen Sulfide"

_genes, 2021, doi:10.3390/genes12030426_

Round 1

Reviewer 1 Report

This is a solid piece of work on an interesting cyanobacterium and the paper is well written. It is a shame that 100% coverage of the genome was not obtained but this is understandable. Analysis of the metabolic potential, electron transport pathways and photoprotection mechanisms needs to be more thorough. More detail is required on some of the pathways discussed in the results. Specific suggestions are outlined below.

Section 3.4- You identified petJ, coding for cytochrome c6, as outlined in table 4. This protein can substitute for plastocyanin and likely performs this step, especially if the environment is copper deficient. Please include this. Were any proteins involved in respiration, i.e. terminal oxidases identified? If so, include details about these since they can play a role in energy generation and as electron acceptors. There are a few reviews on these in the literature, e.g. Mullineaux et al, 2014; Lea-Smith et al, 2016. https://www.ncbi.nlm.nih.gov/pmc/articles/PMC3896814/ https://www.sciencedirect.com/science/article/pii/S0005272815002145

Section 3.5- This section could be more thorough and while it refers to table 5 there is very little information on the respective number of genes in these pathways in this table. Can the authors double check whether this species does encode all the genes in these respective pathways since there are some errors in the Kegg database. Mills et al, 2020 provides a useful overview of these pathways in model cyanobacteria- https://pubmed.ncbi.nlm.nih.gov/32149336/

In addition can mention which essential genes for dissimilatory sulfur metabolism are not present. What are the other three genes necessary for assimilatory nitrate reduction? I thought only nirA and narB were necessary to convert NO3- to NH3?

Section 3.6- Were any genes for surcose biosynthesis identified (i.e. SpsA, Spp)? Sucrose can also act as an osmolyte- https://www.frontiersin.org/articles/10.3389/fmicb.2019.02139/full

As I understand the current literature, the main role of OCP is to allow the cell to cope with high light stress, not UV stress, as implied in the first sentence. It mostly quenches excitation energy in the phycobilisome. There are other proteins involved in coping with high light stress, specifically the flavodiiron proteins, terminal oxidases and the hydrogenase? Are these encoded by this species?

The review of genes encoding proteins synthesising carotenoids is not very thorough. Refer to the Mills et al review mentioned earlier and perform a proper analysis.

Several desaturases are present but only one is mentioned. What are the others?

Reviewer 2 Report

The article presents the draft genome of Phormidium pseudopriestleyi. The organism is interesting as it is capable of sustainining OP at high levels of sulfide in a cold, low light environment. An inventory of key genes has been carried out and some phylogenetic analysis of sqr/D1. The premise of the study is interesting, and the genome assembly appears to have been performed to a good standard.

At present, I find the manuscript is far too long. Although it is clear that considerable work has gone into reviewing the literature surrounding sulfide inhibition of photosynthesis, for this level of detail to be necessary I would expect much deeper characterisation of the mechanisms discussed. This might involve direct comparisons between this and other genomes of other sulfide tolerant cyanobacteria from non-cold environments and/or more lab experiments. My suggestion to the authors is to either carry out this more extensive work or otherwise rework and focus the manuscript and present it as a shorter genome-report style article which would be more appropriate to the level of analysis carried out at this stage.

A few specific points:

  • Please check the literature, I think there are several more Antarctic genomes available now. Either these should be included, or justification for the chosen genomes included
  • BC1401 is from the Arctic
  • Figure 2 is there a better outgroup to use?
  • I would like to see the D1 alignment, perhaps in supplementary. I’m uncertain of the ‘fragments’ and would like to see more to justify that this is a valid approach.
  • I would prefer to see a 16s tree showing P. pseudopriestleyi in context of related sequences
  • There are some confusing aspects of the analysis. For example, the authors state that “All genes necessary for assimilatory sulfur metabolism are present in the MAG, but many essential genes for dissimilatory sulfur metabolism are not.” In table 5, two genes are suggested to be present for dissimilatory sulfur metabolism, while in the Supplementary there is only one. Is there a duplication of Sat? These things should be made clear. I would also be hesitant to describe 2 out of three genes as ‘many’… Unfortunately I do not have the time to check this more extensively, and may have missed something but I urge the authors to double check gene counts etc.

Reviewer 3 Report

Too repetitive Great Oxidation Event, consider changing to GOE instead.

Focus on the discussion about constant light...

Section 1.1.

Total sulfide concentration correspond to the dissolved+particulate fraction? The seawater was prefiltered before quantification?

Please provide reference for the “Standard Methods for the Examination of Water and

Wastewater (21st Edition) from the American Public Health Association” in the references section.

Section 2.2.

Please be more specific on the filtration method (filter type etc...); “Natural History Museum”, where?; provide irradiance/photon flux.

Section 2.3.

Why cite reference [40] if was stated that was extracted according to manufacturer´s instruction, unless there was a slight modification as in [40]?

Section 3.1.

([H2S] + [HS])

Section 3.2.

“Because the genome is incomplete, genes absent from the bin may be present in the organism’s genome”, confusing paragraph.

Phormidium_ median total length (Mb): 4.69_median protein count: 5114

  1. tenue_ median total length (Mb): 5.82_median protein count: 5056
  2. lacuna_ median total length (Mb): 4.81_median protein count: n.d.
  3. ambiguum_ median total length (Mb): 7.41_median protein count: 6262
  4. willie_ median total length (Mb): 4.60_median protein count: 3929

How does the P. pseudopriestleyi is respect to other available genomes for this genus?

Section 3.4.

Would be a good addition a figure with the psbA and psbD cluster compater to other Phormidinium genomes.

Section 4.1.

“A sulfide tolerance mechanism could revolve around replacement and repair of the D1 protein (response 3 or 4 in Fig. 1) or upregulation of the SQR protein (response 2 or 4 in Fig. 1) based on analysis of the MAG”. Too speculative.

Never defined  PAR, instead used photo flux, irradiance... define.

Is confusing, are two (3.4 section) or four (4.1 section) psbA copies in the MAG?

Section 4.2.

Used indiscriminately MAG or P. pseudopriestleyi.

“It is possible that the Lake Fryxell P. pseudopriestleyi has lost these genes due to its

low light environment, in which case, photoprotective and UV exposure genes may be

present in closely related organisms found in environments with higher light stress.” Provide reference.

Section 4.3.

“P. pseudopriestleyi is present in both locations suggesting it is widespread in Antarctica and adapted to varying levels of salinity and UV stress in addition to sulfide. This suggests

that P. pseudopriestleyi is not only able to grow in a sulfidic, cold and low-light aquatic

environment, but also under elevated UV and hypersaline conditions, and may possess

photoprotective genes in these environments.”

Based on two ponds and one lake, these paragraph is too speculative.

Despite the high number of methane-related genes, there is no discussion about it.

Round 2

Reviewer 3 Report

A minor comment.

"Point 4: (Section 2.2) Please be more specific on the filtration method (filter type etc...); “Natural History Museum”, where?; provide irradiance/photon flux.

Response 4: The lake water was filtered through a sterile syringe with a 0.2 um syringe filter. The Natural History Museum where the culture was grown is in London, UK, and the culture was grown in the presence of 500 lux. This information has been added to section 2.2."

The current unit for [light] in PAR is µmol quanta m-2 s-1.

Author Response

Thank you for the the comment. The lux units have been updated to umol photons m-2 s-1.